# Deferred Backdoor Functionality Attacks on Deep Learning Models

## Abstract

Deep learning models are vulnerable to backdoor attacks, where adversaries inject malicious functionality during training that activates on trigger inputs at inference time. Extensive research has focused on developing stealthy backdoor attacks to evade detection and defense mechanisms. However, these approaches still have limitations that leave the door open for detection and mitigation due to their inherent design to cause malicious behavior in the presence of a trigger. To address this limitation, we introduce Deferred Activated Backdoor Functionality (DABF), a new paradigm in backdoor attacks. Unlike conventional attacks, DABF initially conceals its backdoor, producing benign outputs even when triggered. This stealthy behavior allows DABF to bypass multiple detection and defense methods, remaining undetected during initial inspections. The backdoor functionality is strategically activated only after the model undergoes subsequent updates, such as retraining on benign data. DABF attacks exploit the common practice in the life cycle of machine learning models to perform model updates and fine-tuning after initial deployment. To implement DABF attacks, we approach the problem by making the unlearning of the backdoor fragile, allowing it to be easily cancelled and subsequently reactivate the backdoor functionality. To achieve this, we propose a novel two-stage training scheme, called `DeferBad`. Our extensive experiments across various fine-tuning scenarios, backdoor attack types, datasets, and model architectures demonstrate the effectiveness and stealthiness of `DeferBad`.

## 1 Introduction

Deep neural networks (DNNs) have achieved remarkable performance across various application domains, revolutionizing fields such as computer vision, natural language processing, and robotics. However, their complex, opaque nature leaves them vulnerable to exploitation. One particularly concerning vulnerability is backdoor attacks, where an adversary injects malicious functionality into a model during training that remains hidden until activated by a trigger pattern in inputs at inference time (Gu et al., 2017; Liu et al., 2018b). Backdoors enable targeted misclassification of inputs with the trigger to a desired label, while the model behaves normally on clean inputs. This makes backdoors hard to detect and a serious threat, especially if the model is deployed in safety-critical applications.

Extensive research has focused on developing increasingly sophisticated and stealthy backdoor attacks to evade defense mechanisms (Chen et al., 2017; Nguyen & Tran, 2020; Li et al., 2021b). These approaches have significantly enhanced the covertness of backdoors, making them more challenging to identify and mitigate. However, despite these advancement, current backdoor techniques remain constrained by a *fundamental limitation*: the inherent necessity of activating backdoor functionality. This core characteristic to trigger malicious behaviors for attack's successes paradoxically renders the backdoor weak at detection and mitigation in defense stages. For instance, a careful analysis through reverse engineering techniques targeting specific output classes can potentially uncover the presence of a backdoor (Wang et al., 2019). Additionally, methods leveraging the model's output patterns have shown promise in identifying backdoored models (Gao et al., 2019). Thus, the crucial feature that triggers backdoor attacks also serves as its Achilles' heel by providing avenues toward potential detection and mitigation.

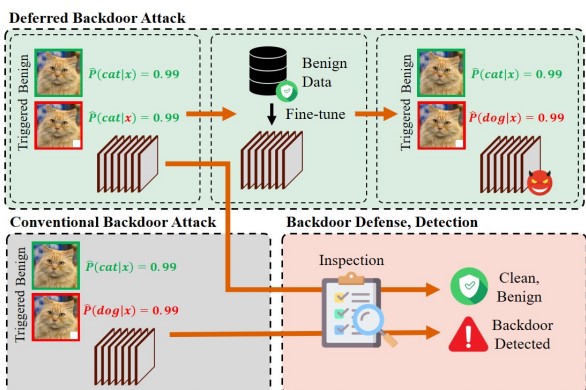

Figure 1: An illustrating example of backdoor attacks.

To overcome this fundamental limitation, we introduce a novel attack strategy: Deferred Activated Backdoor Functionality (DABF). This concept represents a significant shift in backdoor attack approaches, as it allows the backdoor to remain *dormant* in deployed models, even in the presence of the trigger. In particular, DABF consists of two phases: a backdoor *dormancy phase* and a backdoor *deferred activation phase*. In the dormancy phase, the compromised model behaves indistinguishably from a clean model when deployed, making it much harder to detect. Later in the deferred activation phase, the backdoor functionality is activated when the model is fine-tuned on a benign dataset, without any further involvement of attackers.

One important feature by DABF in the backdoor *dormancy phase* is that DABF fundamentally challenges the assumptions of current defense mechanisms (e.g., detection via reverse engineering techniques on specific output classes (Wang et al., 2019; Guo et al., 2019; Wang et al., 2022a) or via analyzing model's output patterns (Gao et al., 2019; Guo et al., 2023; Hou et al., 2024)). By keeping the backdoor dormant until activation, DABF can potentially bypass not only existing defenses but also future approaches that rely on similar assumptions. Moreover, DABF presents a unique advantage: it can potentially evade detection even in stronger scenarios, i.e., a defender knows the trigger, where all previous backdoor attacks fail. This capability represents a new level of stealth in backdoor attacks, significantly raising the bar for detection and mitigation strategies. Another important feature of DABF is that the dormant backdoor is activated without any intervention by attackers. In particular, DABF exploits the common scenario where a deployed model is thoroughly inspected and deemed clean but then retrained with additional data. This situation frequently arises in practice when a model is updated to improve performance, adapted to new data distributions, or learned new tasks (Wang et al., 2024). The model owner may collect additional training data over time and fine-tune the model, unaware that this process could activate a hidden backdoor.

Furthermore, DABF offers an additional layer of protection for the attacker. Even if continuous monitoring eventually detects the backdoor after its activation, the attacker can plausibly avoid suspicion. This is because at the time of the model's deployment and initial security checks, no backdoor was detectable. The backdoor's activation occurs solely due to the routine actions of the model's owners or users, without any further intervention from the attacker. This temporal disconnect between the attacker's actions and the backdoor's activation makes it extremely challenging to attribute the backdoor to any specific individual or action.

To achieve DABF, we propose `DeferBad`, leveraging the key insight that neural networks have an inherent tendency to rediscover suppressed behavior during benign retraining (Qi et al., 2023). This novel two-phase approach consisting of an initial backdoor injection phase followed by a strategic partial model update for concealment. Our method selectively updates a subset of the model's layers during the concealment phase, creating an unstable equilibrium in the network. This carefully crafted state is designed to be easily disrupted by subsequent fine-tuning, regardless of the specific fine-tuning strategy employed, establishing a comprehensive method that effectively ensures covert backdoor reactivation.

Our main contributions are as follows:

| Feature | Conventional (Gu et al., 2017) | Latent (Yao et al., 2019) | UBA-Inf (Huang et al., 2024) | DeferBad (ours) |
|---|---|---|---|---|
| Deferred backdoor | ✗ | ✓ | ✓ | ✓ |
| Normal behavior w/ trigger | ✗ | ✗ | ✓ | ✓ |
| Attacker's intervention | ✗ | ✓ | ✗ | ✓ |
| Activation mechanism | - | fine-tuning $k \ll$ (# all layers) | unlearning | fine-tuning $k \leq$ (# all layers) |

Table 1: Comparison of related papers and the proposed DeferBad.

- We propose Deferred Activated Backdoor Functionality (DABF), a novel approach designed to fundamentally bypass existing backdoor detection methods. To the best of our knowledge, DABF is the first method to temporarily conceal the very functionality that defines a backdoor and reactivate it afterward. Consequently, our approach offers the potential to evade detection even in scenarios where defenders have knowledge of the trigger, where all previous backdoor attacks fail.

- We introduce DeferBad, a specific implementation of DABF. DeferBad effectively conceals backdoors and ensures their reactivation in line with DABF principles. Importantly, DeferBad demonstrates robustness across various fine-tuning scenarios and backdoor trigger types, showcasing its versatility and general applicability.

- We empirically evaluate the effectiveness of DeferBad across two datasets (i.e., CIFAR10, TinyImageNet), three model architectures (i.e., ResNet18, VGG16, and EfficientNet-B0), and two backdoor attack types (i.e., BadNets, ISSBA). We explore various fine-tuning scenarios, including different numbers of updated layers and distribution shift retraining (using CIFAR10-C and TinyImageNet-C). Furthermore, we analyze DeferBad's stealthiness against seven state-of-the-art backdoor detection and mitigation methods (i.e., Neural Cleanse, STRIP, Fine-Pruning, GradCAM, RCS, Scale-Up, and IBD-PSC).

Our work not only presents a novel attack strategy but also reveals critical vulnerabilities in current machine learning practices, emphasizing the need for continuous security measures throughout a model's lifecycle.

## 2 RELATED WORK

### 2.1 BACKDOOR ATTACKS

Backdoor attacks in deep neural networks (DNNs) have emerged as a significant security concern, particularly in image processing applications. Gu et al. (2017) demonstrated DNNs' vulnerability to such attacks and proposed BadNets, which injects backdoors by poisoning training data with specific trigger patterns. Following this, research has focused on enhancing the stealth of backdoor attacks through various trigger designs. Chen et al. (2017) employed a blended strategy for more covert triggers, while Nguyen & Tran (2020) developed input-aware dynamic triggers. Li et al. (2021b); Doan et al. (2021); Wang et al. (2022b) further advanced stealth by creating invisible, sample-specific backdoor triggers. Additionally, clean label poisoning methods (Turner et al., 2019; Saha et al., 2020; Zeng et al., 2023) have been explored to make backdoor attacks even more difficult to detect during the training process. Recent works Chen et al. (2022); Jha et al. (2023) have shown that backdoors can be injected using only clean images with poisoned labels, further enhancing the stealthiness of the attack. These advancements in backdoor attack techniques have predominantly focused on scenarios where the backdoor functionality is immediately activated upon the model's deployment, leaving a gap in understanding delayed activation mechanisms.

The concept of deferred backdoor activation has been explored in different ways. Yao et al. (2019) proposed latent backdoors that implant backdoors in the latent representation of pre-trained models without including the target class. These backdoors remain dormant in the pre-trained model and activate only when fine-tuned on a dataset with the target class. However, these latent backdoors do not maintain normal behavior in the presence of triggers during the dormant phase, as they pro-

duce significantly different latent representations for triggered inputs. Moreover, their effectiveness diminishes as more layers are fine-tuned.

More recently, several studies have explored unlearning-based deferred backdoor attacks (Di et al., 2022; Liu et al., 2024; Huang et al., 2024). These approaches implement deferred backdoor attacks using unlearning as their activation mechanism. While these approaches maintain normal behavior with triggers during the dormant phase, their practical applicability is limited due to the restricted availability of unlearning services and the requirement for attacker intervention in the activation process. In contrast, as shown in Table 1, `DeferBad` addresses these limitations by leveraging commonly used fine-tuning processes for backdoor activation. This approach is particularly practical as fine-tuning is ubiquitously supported across major deep learning frameworks, requires no attacker intervention, and remains effective regardless of which layers are fine-tuned.

## 2.2 BACKDOOR DEFENSES

Numerous techniques have been developed to detect and mitigate against backdoor attacks in deep neural networks. These methods can be broadly categorized into detection and mitigation strategies.

Detection methods aim to identify the presence of backdoors in trained models or input data. STRIP (Gao et al., 2019) detects whether an input contains a strong backdoor trigger by analyzing the model's output entropy under input perturbations. Activation Clustering (Chen et al., 2018) identifies anomalous activation patterns caused by backdoors in the neural network's intermediate layers. Spectral Signatures (Tran et al., 2018) leverages singular value decomposition to identify a concentrated distribution of backdoored training samples. SentiNet (Chou et al., 2020) utilizes GradCAM (Selvaraju et al., 2017) to identify trigger regions in input images and detect potential backdoors. Random Channel Shuffling (RCS) (Cai et al., 2022) exploits the observation that trigger information tends to be concentrated in specific channels by analyzing class-wise variations under channel perturbations. Scale-Up (Guo et al., 2023) examines prediction consistency under image amplification to detect backdoors. IDB-PSC (Hou et al., 2024) analyzes the model's behavior under batch normalization parameter scaling to identify potential backdoors. Other defense strategies, on the other hand, focus on mitigating or removing backdoors from compromised models. Neural Cleanse (Wang et al., 2019) uses optimization techniques to reverse engineer potential triggers and subsequently remove them. Fine-pruning (Liu et al., 2018a) aims to eliminate neurons that are unimportant for clean data, thereby weakening the backdoor without significantly affecting the model's primary task performance. Neural Attention Distillation (NAD) (Li et al., 2021a) employs model distillation to transfer knowledge from a clean teacher model to remove backdoors. CLP (Zheng et al., 2022) detects and eliminates trigger-sensitive channels in a data-free manner.

However, it is crucial to note that many of these detection and defense techniques operate under the assumption that backdoored models will exhibit anomalous behavior in the presence of trigger inputs (Gao et al., 2019; Wang et al., 2019; Chou et al., 2020; Guo et al., 2023). This fundamental assumption limits their effectiveness against DABF attack that do not immediately activate upon deployment. Moreover, while knowing the backdoor trigger can significantly enhance detection and mitigation capabilities, it often provides an unrealistic advantage to defenders. In contrast, our proposed DABF challenges this paradigm. Even with knowledge of the trigger, DABF can potentially evade detection methods as it remains dormant until activated through fine-tuning, presenting a novel challenge to existing backdoor defense strategies.

## 3 THREAT MODEL: DEFERRED BACKDOOR ATTACK

We propose a novel threat model centered on a Deferred Activated Backdoor Functionality (DABF) attack, which represents a significant evolution in the landscape of adversarial machine learning. This attack exploits the common practice of fine-tuning in the deep learning model lifecycle, presenting unique challenges to current security paradigms. In the DABF attack scenario, an adversary crafts a model with a latent backdoor that *remains dormant* during initial deployment but *activates* upon fine-tuning with clean data. This approach differs fundamentally from traditional backdoor attacks in two critical aspects:

- Initial dormancy: The backdoor remains inactive during post-deployment, with the model exhibiting normal behavior on all inputs, including those containing triggers.

- Deferred activation: The backdoor activates automatically during fine-tuning on clean data, without further adversarial intervention.

The attack targets the fine-tuning stage of the deep learning lifecycle, which typically follows initial training and deployment. This stage, crucial for transfer learning and domain adaptation, inadvertently serves as the activation mechanism for the latent backdoor. The adversary's capabilities are limited to the initial training phase, with no access or influence during the subsequent fine-tuning process. We formalize the DABF attack as an optimization problem: Let $f \in \mathcal{F}$ be the backdoored model, $L_{01}(\cdot, \cdot)$ be the classification error, $T(x)$ be the backdoor-trigger injection function, $\eta(y)$ be the target label function, and $g = \text{ft}(f, \mathcal{D})$ be the fine-tuned model derived from $f$ using a dataset $\mathcal{D}$ for fine-tuning. The objective is defined as:

$$\min_f \overbrace{\mathbb{E}_{(x,y)\sim D}[L_{01}(g(T(x)), \eta(y))]}^{\text{(i)}} + \overbrace{\mathbb{E}_{(x,y)\sim D}[L_{01}(g(x), y)]}^{\text{(ii)}} \tag{1}$$

$$\text{subj. to} \quad \underbrace{\mathbb{E}_{(x,y)\sim D}[L_{01}(f(T(x)), \eta(y))] \geq 1 - \epsilon}_{\text{(iii)}}, \quad \underbrace{\mathbb{E}_{(x,y)\sim D}[L_{01}(f(x), y)] \leq \epsilon'}_{\text{(iv)}}.$$

Here, the objective is finding an initial model $f$, if it is finetuned, i.e., $g$, an implemented backdoor is activated, i.e., small (i), while the finetuned model is still performant on normal data, i.e., small (ii). But, the constraints ensure that the initial model $f$ should not trigger backdoors, i.e., satisfying (iii), but is still performant on clean data, i.e., satisfying (iv), to effectively conceal the backdoor in the pre-fine-tuning stage for some small $\epsilon$ and $\epsilon'$.

## 4 METHODOLOGY: DEFERBAD

This section presents our approach to creating a Deferred Activated Backdoor Functionality (DABF). We first provide the intuition behind our method, followed by a detailed description of the implementation.

### 4.1 INTUITION

Our approach is inspired by observations in machine learning, particularly in the context of safety alignment in Large Language Models (LLMs) and backdoor learning. It has been observed that after safety alignment training, subsequent fine-tuning on general data often results in a partial degradation of the safety measures (Qi et al., 2023). This phenomenon aligns with our observations in backdoor learning, where after a typical cycle of backdoor *learning* followed by backdoor *unlearning* (generally achieved through parameter updates), subsequent fine-tuning often resulted in a partial reactivation of the backdoor, i.e., $\mathbb{E}_{(x,y)\sim D}[L(g(T(x)), \eta(y))]$, is reduced. This heuristically achieves the goal of attackers in (1).

Based on these insights, we hypothesized that if we could design a method to effectively counteract backdoor unlearning when optimized on clean data, we could achieve our objective of creating a deferred backdoor activation. This hypothesis led us to formulate a key question: How can we structure the initial model such that fine-tuning on clean data effectively cancels out the backdoor unlearning process? To address this challenge, we developed a novel two-phase method: backdoor injection followed by partial model update for concealment.

### 4.2 METHOD

Our method consists of two main steps: backdoor injection and partial concealment.

**Backdoor Injection:** We first train the model on a poisoned dataset $\mathcal{D}_{\text{poison}}$, defined as:

$$\mathcal{D}_{\text{poison}} = \{(T(x), \eta(y)) \text{ with probability } p, \text{ else } (x, y) \mid (x, y) \in \mathcal{D}\}, \tag{2}$$

where $p$ is the poison rate, and $\mathcal{D}$ is the clean dataset.

**Backdoor Concealment:** After injecting the backdoor, We then perform a partial update of the model to conceal the backdoor. This is done using an unlearning dataset $\mathcal{D}_{\text{unlearn}}$:

$$\mathcal{D}_{\text{unlearn}} = \{(T(x), y) \text{ with probability } p, \text{ else } (x, y) \mid (x, y) \in \mathcal{D}\} \tag{3}$$

Crucially, we update a subset of the model's layers, denoted by $\theta_{\text{update}}$, according to:

$$\theta'_{\text{update}} = \theta_{\text{update}} - \alpha \nabla_{\theta_{\text{update}}} \mathbb{E}_{(x,y) \sim \mathcal{D}_{\text{unlearn}}} [L(f_\theta(x), y)] \tag{4}$$

where $\alpha$ is the learning rate and $L$ is the convex loss function of the classification error $L_{01}$.

---

**Algorithm 1** DeferBad: Attacker's Algorithm

---

**Require:** Dataset $D$, Model $M$, Trigger function $T$, Target label $y_t$
**Ensure:** Backdoored model $M_b$
  1: $M_b \leftarrow \text{BackdoorInjection}(M, D, T, y_t)$                 ▷ See Table 2
  2: $M_b \leftarrow \text{BackdoorConcealment}(M_b, D, T)$           ▷ See Table 2
  3: **return** $M_b$

---

**Algorithm 2** User's Fine-tuning Algorithm

---

**Require:** Backdoored model $M_b$, Fine-tuning dataset $D_f$
**Ensure:** Fine-tuned model $M_f$
  1: $M_f \leftarrow M_b$
  2: Train $M_f$ on $D_f$ according to user's preferences        ▷ See Table 2
  3: **return** $M_f$

---

The choice of which layers to update (i.e., $\theta_{\text{update}}$) is carefully designed based on the model's architecture, with particular attention to the presence or absence of batch normalization (BN) layers. This distinction is crucial because BN layers significantly influence the model's behavior during fine-tuning, which is key to our backdoor activation mechanism.

For models without BN, we update the last $k$ layers, setting $\theta_{\text{update}} = \theta_{\text{last-}k}$. This approach creates a temporary equilibrium where the modified last layers compensate for the backdoor behavior of the earlier layers, effectively concealing the backdoor. By concentrating our concealment efforts in these final layers, we address the common practice of fine-tuning only the last few layers of a pre-trained model, which is often done to save computational resources or prevent overfitting. When such partial fine-tuning occurs, it directly impacts these carefully calibrated layers, easily disrupting the concealment and reactivating the backdoor. This method also works effectively in a full-fine-tuning scenario. When all layers are updated during fine-tuning, the earlier layers, which still contain latent backdoor information, are optimized alongside the last layers. This simultaneous optimization creates a synergistic effect: as the earlier layers evolve, they push the model towards rediscovering the backdoor pattern, while the changes in the last layers further destabilize the concealment state. This dual movement significantly contributes to backdoor reactivation, leveraging the model's inherent tendency to rediscover suppressed patterns during retraining.

For models with BN, we update the first $k$ layers ($\theta_{\text{update}} = \theta_{\text{first-}k}$) while disabling BN statistic updates, instead using running averages. This approach exploits BN layers' sensitivity to distribution shifts. By modifying early layers and freezing BN statistics, we create a scenario where fine-tuning, whether partial or full, causes significant distribution shifts in BN layers, triggering backdoor reactivation. Specifically, unlearning the first layers suppresses backdoor activations without completely eliminating them. During subsequent fine-tuning, as BN layers adapt, they amplify these suppressed activations, effectively reactivating the backdoor. This method is robust across various fine-tuning scenarios, including partial updates, full fine-tuning, or even cases where only BN statistics are updated.

## 5 EXPERIMENTS

In this section, we evaluate `DeferBad` from different perspectives. We first present the experiment setup in Section 5.1. In Section 5.2, we show the effectiveness in term of backdoor dormancy and

Table 2: Comprehensive Experiment Settings and Hyperparameters

| Stage | Parameter | Value (BN models) | Value (non-BN models) |
|---|---|---|---|
| Backdoor Injection | Poisoning Rate | 10% | 10% |
| | Epochs | 100 | 100 |
| | Optimizer | SGD with cosine annealing | SGD with cosine annealing |
| | Learning Rate | 0.001 | 0.001 |
| Backdoor Concealment | Poisoning Rate | 50% | 10% |
| | Optimizer | Adam | Adam |
| | Learning Rate | 0.0001 | 0.0001 |
| | ASR Threshold | Empirically determined | Empirically determined |
| | BN Update | Disabled | N/A |
| | Layers to Update | First $k$ layers | Last $k$ layers |
| | Layers to Freeze | Last $(n-k)$ layers | First $(n-k)$ layers |
| User Fine-tuning | Layers to Update | Last $k$ layers, $k \leq n$ (user-defined) | |
| | Learning Rate | $\alpha$ (user-defined) | |
| | Epochs | $E$ (user-defined) | |
| | BN Behavior | Default (enabled) | |
| | Optimizer | User's choice | |

activation after fine-tuning. Then, we evaluate `DeferBad`'s resistance to existing defenses during the dormancy phase in Section 5.3.

## 5.1 EXPERIMENTAL SETUP

We evaluate `DeferBad` on CIFAR-10 (Krizhevsky & Hinton, 2009) and Tiny ImageNet (Li, 2015) datasets. CIFAR-10 contains 50,000 training and 10,000 test images of size 32x32 in 10 classes, while Tiny ImageNet has 100,000 training and 10,000 test images of size 64x64 in 200 classes. For both datasets, we further split the test set into 5,000 validation and 5,000 test images to ensure robust evaluation. We experiment with three DNN architectures: ResNet18 (He et al., 2016), VGG16 (Simonyan & Zisserman, 2014), and EfficientNet-B0(Tan, 2019). To explore various backdoor triggers, we implemented both BadNets (Gu et al., 2017) and ISSBA (Li et al., 2021b). For BadNets, we used a 3x3 pixel pattern trigger for CIFAR-10 and a 6x6 pixel pattern trigger for Tiny ImageNet, while ISSBA employed a StegaStamp encoder with a 100-bit secret.

Our experimental procedure follows three main stages as outlined in Table 2: Backdoor Injection, Backdoor Concealment, and User Fine-tuning. For the Backdoor Injection stage, we first train the model benignly, then inject the backdoor using the parameters specified in the table. The Backdoor Concealment stage employs different strategies based on the model architecture, particularly differentiating between models with and without batch normalization (BN) layers.

For fine-tuning, we explore two scenarios:

1. Retraining on new data from a similar distribution by excluding 5,000 images from the training set during the initial stages and including them during fine-tuning.

2. Fine-tuning on different distributions using corruption datasets CIFAR10-C (Hendrycks & Dietterich, 2019), applying fog, noise, and JPEG compression corruptions at severity levels 1, 3, and 5.

Overall, we set k to 4, freezing the corresponding 4 convolutional layers, and then performed fine-tuning. detailed information about the hyperparameters, optimization strategies, and specific settings for each stage and model type, please refer to Table 2. All experiments were conducted on a single RTX 3090 GPU.

### 5.1.1 EVALUATION SETUP

To evaluate the stealthiness and effectiveness of `DeferBad`, we measure the clean accuracy (CA) and attack success rate (ASR) of the backdoored model at each stage of the attack pipeline. CA is the classification accuracy on clean test inputs, while ASR is the fraction of triggered test inputs that are misclassified into the attacker's target class. A successful `DeferBad` model should have high CA and low ASR after backdoor concealment to evade detection, but high ASR after fine-tuning to be effective.

Table 3: Results for all stages of `DeferBad` using different attack types on CIFAR-10. Values represent $^{\text{Clean Accuracy (CA)}}/_{\text{Attack Success Rate (ASR)}}$ in percentage.

| Model | Attack | 1: Injection (↑)/(↑) | 2: Concealment (↑)/(↓) | 3: After FT (↑)/(↑) |
|---|---|---|---|---|
| ResNet18 | BadNet | 95.26 / 97.09 | 94.90 / 0.07 | 95.28 / 94.07 |
| | ISSBA | 95.16 / 99.98 | 94.54 / 0.27 | 95.08 / 84.65 |
| VGG16 | BadNet | 91.24 / 96.65 | 90.10 / 0.04 | 91.60 / 93.23 |
| | ISSBA | 91.22 / 99.69 | 91.20 / 0.60 | 91.62 / 48.54 |
| EfficientNet-B0 | BadNet | 91.36 / 97.35 | 91.48 / 0.49 | 90.66 / 86.13 |
| | ISSBA | 91.10 / 99.80 | 91.04 / 0.38 | 89.82 / 58.17 |

Table 4: Clean Accuracy (CA) and Attack Success Rate (ASR) for different models and attack types on CIFAR10-C dataset (JPEG compression) across severities, before and after fine-tuning (FT). Values represent $^{\text{Clean Accuracy (CA)}}/_{\text{Attack Success Rate (ASR)}}$ in percentage.

| Model | Attack | Severity 1 | | Severity 3 | | Severity 5 | |
|---|---|---|---|---|---|---|---|
| | | Before FT (↑)/(↑) | After FT (↑)/(↓) | Before FT (↑)/(↑) | After FT (↑)/(↑) | Before FT (↑)/(↑) | After FT (↑)/(↓) |
| ResNet18 | BadNet | 87.06 / 0.19 | 90.05 / **94.28** | 80.48 / 0.52 | 84.40 / 91.53 | 75.59 / 0.80 | 79.84 / 76.67 |
| | ISSBA | 87.00 / 0.70 | 89.69 / 80.28 | 79.36 / 0.65 | 84.59 / 72.18 | 73.15 / 1.14 | 70.13 / 75.30 |
| VGG16 | BadNet | 83.83 / 0.0 | 86.70 / **98.34** | 78.44 / 0.0 | 83.41 / **97.90** | 74.21 / 0.0 | 80.03 / **84.45** |
| | ISSBA | 85.64 / 0.98 | 85.99 / **97.98** | 80.93 / 1.29 | 82.35 / **98.91** | 77.51 / 1.62 | 78.98 / **98.95** |
| EfficientNet-B0 | BadNet | 83.38 / 0.66 | 83.64 / 59.52 | 76.81 / 0.98 | 77.55 / 45.93 | 71.49 / 0.91 | 73.88 / 42.36 |
| | ISSBA | 83.29 / 0.36 | 83.15 / **61.68** | 76.43 / 0.57 | 77.28 / **59.18** | 71.64 / 0.61 | 72.71 / **68.15** |

## 5.2 Effectiveness on Backdoor Injection, Concealment, and Reactivation

We evaluate the effectiveness of our `DeferBad` approach across different model architectures, attack types, and datasets. Table 3 presents the results for CIFAR-10, showing Clean Accuracy (CA) and Attack Success Rate (ASR) for each stage of our attack.

Our results demonstrate that `DeferBad` successfully conceals backdoors to near-undetectable levels while achieving significant ASR after fine-tuning across all tested scenarios. We observe that after the concealment stage, the ASR drops to near-zero levels (0.07% - 0.60%), effectively hiding the backdoor. Crucially, after fine-tuning, the ASR significantly increases, reaching 94.07% for ResNet18 with BadNet, 93.23% for VGG16 with BadNet, and 97.35% for EfficientNet with BadNet, while maintaining or increasing high clean accuracy. This confirms the success of our deferred activation mechanism. ISSBA attacks show lower but still significant ASR after fine-tuning (84.65% for ResNet18, 48.54% for VGG16, and 61.68% for EfficientNet), suggesting that more complex triggers might be slightly more challenging to reactivate but still remain highly effective.

We further tested our approach under distribution shift scenarios using CIFAR10-C, as shown in Table 4. The results for JPEG compression at different severity levels reveal that our backdoor remains effective even under data distribution changes. Notably, in some cases (highlighted in bold), the ASR under distribution shift is even higher than in the original distribution, particularly for VGG16. This unexpected behavior suggests that our backdoor might be leveraging certain robustness properties of the model, an intriguing area for future investigation.

Our experiments with varying numbers of fine-tuned layers (Fig. 2) reveal interesting trends. Generally, ASR tends to increase when fewer layers are fine-tuned. For models with BN (e.g., EfficientNet), even minimal layer updates provide sufficient conditions for reactivation, while updating more layers can interfere with this process. For models without BN (e.g., VGG16), ASR is highest when fine-tuning focuses on the last few layers where reactivation-related features are concentrated, with additional layer updates potentially disrupting these patterns. However, fine-tuning more layers, especially in VGG16 ISSBA and EfficientNet, occasionally resulted in ASR dropping below 10%. Despite this, most scenarios maintained significant ASR. Notably, VGG16 showed lower performance when only the layer used for unlearning was fine-tuned, suggesting that fine-tuning preceding layers helps align with the concealed layer. Overall, these results demonstrate that `DeferBad` remains effective across various fine-tuning strategies, highlighting its robustness and versatility as an attack vector. Further results for Tiny ImageNet and additional corruption types are presented in Appendix A, B, showing consistent performance across different datasets and perturbation types.

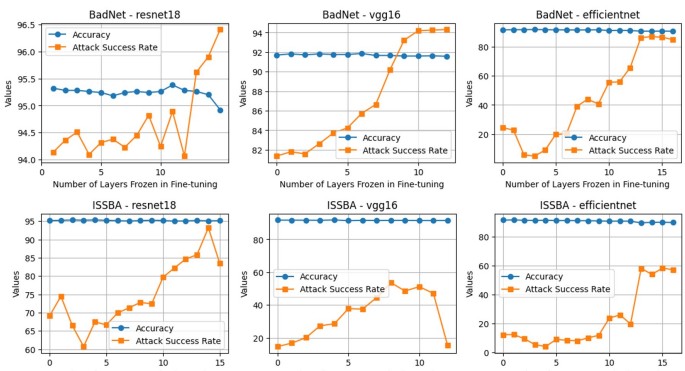

Figure 2: Impact of the number of fine-tuned layers on Clean Accuracy (CA) and Attack Success Rate (ASR) for ResNet18 on CIFAR-10.

### 5.3 STEALTHINESS

To evaluate the stealthiness of `DeferBad`, we tested it against seven state-of-the-art backdoor detection and mitigation methods: Neural Cleanse (Wang et al., 2019), STRIP (Gao et al., 2019), GradCAM (Selvaraju et al., 2017), Fine-Pruning (FP) (Liu et al., 2018a), Random Channel Shuffling (RCS) (Cai et al., 2022), Scale-Up (Guo et al., 2023), and IDB-PSC (Hou et al., 2024). We conducted experiments on ResNet18, using Badnet, which was detectable by all methods when injected using conventional techniques.

**Neural Cleanse:** `DeferBad` fundamentally evades detection by Neural Cleanse. As shown in Figure 3b, the anomaly index for `DeferBad`-infected models (0.672) was even lower than that of clean models (0.778) on CIFAR-10, while similar trends were observed on Tiny-ImageNet (1.796 vs 1.220). In both cases, BadNet models showed significantly higher anomaly indices (4.02 and 3.549 respectively). This result demonstrates `DeferBad` is resilient to Neural Cleanse as expected.

**STRIP:** Similarly, STRIP fails to detect `DeferBad` because the trigger does not alter the model's output before backdoor activation. Figure 3d demonstrates that the entropy distribution for `DeferBad` models was actually higher than that of normal models. Given that lower entropy is typically associated with a higher likelihood of a backdoor, this result further demonstrates `DeferBad`'s ability to evade detection.

**GradCAM:** Our analysis using GradCAM, as illustrated in Figure 3a, revealed minimal difference in the activation maps between clean inputs and triggered inputs for `DeferBad` models. While backdoor models show distinct attention patterns focused on the trigger area, `DeferBad` models exhibit saliency maps very similar to clean models. This similarity in model attention further underscores the stealthy nature of `DeferBad`, as it does not introduce easily detectable changes in the model's decision-making process. Consequently, `DeferBad` is likely to evade detection methods that rely on visual explanations, such as SentiNet (Chou et al., 2020) and Februus (Doan et al., 2020). Note that GradCAM is only used for qualitative measures for inspecting backdoors (Li et al., 2021b; Doan et al., 2021)

**Fine-Pruning (FP):** We evaluated FP's effectiveness in mitigating `DeferBad` by fine-tuning models after the fine-pruning process across different datasets. Our results reveal dataset-dependent patterns in the defense's effectiveness. On CIFAR-10, as shown in Figure 3c, FP was only partially effective: ASR remained relatively stable around 40% after fine-tuning, regardless of the pruning level, while clean accuracy decreased with increased pruning. However, experiments on Tiny ImageNet showed markedly different results. When fine-tuning the pruned models, FP proved to be highly effective on this dataset, with ASR dropping to nearly 0% as pruning progressed. This contrast in effectiveness suggests that the resilience of `DeferBad` against pruning-based defenses varies significantly depending on the dataset complexity.

We conducted additional experiments with three recent detection methods: RCS (Cai et al., 2022), Scale-Up (Guo et al., 2023), and IDB-PSC (Hou et al., 2024). While RCS showed some capability in detecting `DeferBad`, the detection scores were significantly lower compared to conventional

BadNet attacks. Scale-Up and IDB-PSC were effectively evaded by `DeferBad`. Detailed results for these additional experiments are presented in Appendix D.

These results demonstrate that while `DeferBad` may not completely evade all detection methods, it significantly reduces detection signals compared to conventional backdoor attacks. By fundamentally changing how the backdoor manifests in the model, `DeferBad` shows improved stealthiness against most detection methods.

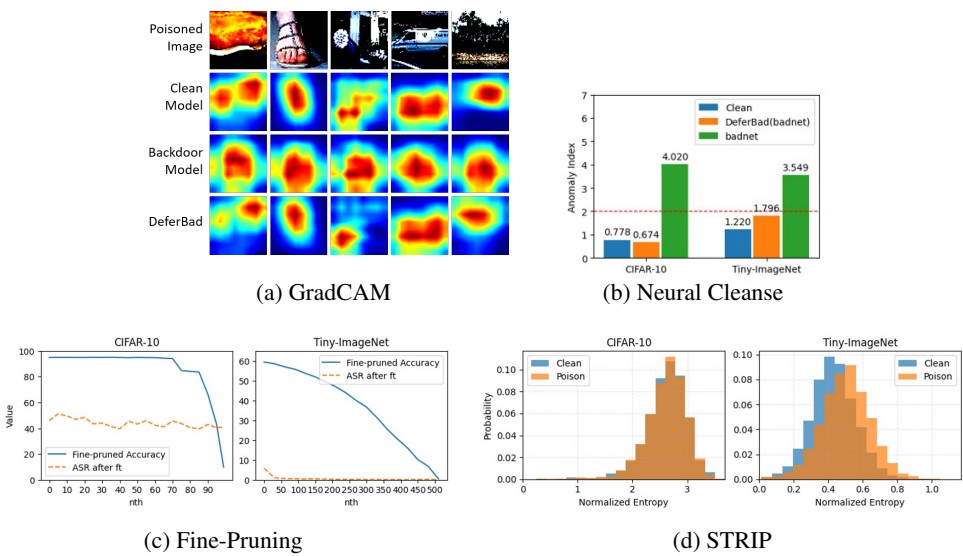

Figure 3: Results of various backdoor detection techniques applied to our DABF model. (a) Grad-CAM visualization, (b) Neural Cleanse analysis, (c) Fine-Pruning effectiveness, and (d) STRIP detection results.

## 6 CONCLUSION

In this paper, we introduced Deferred Activated Backdoor Functionality (DABF), a novel backdoor attack strategy that fundamentally challenges current approaches to AI security. DABF addresses the key limitation of existing backdoor techniques by keeping the backdoor dormant during the initial deployment phase and activating it through routine model updates like fine-tuning. Our implementation, `DeferBad`, has demonstrated remarkable effectiveness across various datasets, model architectures, and attack scenarios. Key achievements of `DeferBad` include successful concealment of backdoors during initial deployment, significant attack success rates after fine-tuning while maintaining competitive clean accuracy, and robustness against various fine-tuning strategies and distribution shifts. Notably, `DeferBad` has shown the ability to bypass multiple state-of-the-art backdoor detection and mitigation techniques. Our work underscores critical vulnerabilities in the lifecycle management of AI models, emphasizing that the absence of immediate backdoor indicators does not guarantee long-term security. This finding calls for a paradigm shift in AI security practices, necessitating the development of continuous and evolving detection methods throughout a model's operational life. However, our research also has limitations. The current study focuses exclusively on vision tasks, and the effectiveness of DABF in other domains, such as natural language processing or speech recognition, remains to be explored. Looking ahead, it would be interesting to investigate the applicability of DABF to other AI domains and explore its interaction with different model architectures and learning paradigms. Furthermore, It would also be intriguing to examine how DABF performs not only under fine-tuning scenarios but also with other model update techniques such as pruning, quantization, or knowledge distillation. These investigations could further our understanding of the vulnerabilities and resilience of AI models throughout their lifecycle.

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

## A    RESULTS ON TINY IMAGENET

Table 5 presents the results for all stages of DABF using different attack types on Tiny ImageNet. The table shows Clean Accuracy (CA) and Attack Success Rate (ASR) for various models and attack types across different stages of the DABF process.

Table 5: Results for all stages of DABF using different attack types on Tiny ImageNet. Values represent $^{\text{Clean Accuracy (CA)}}/_{\text{Attack Success Rate (ASR)}}$ in percentage.

| Model | Attack | 1: Injection $^{(\uparrow)}/_{(\uparrow)}$ | 2: Concealment $^{(\uparrow)}/_{(\downarrow)}$ | 3: After FT $^{(\uparrow)}/_{(\uparrow)}$ |
|---|---|---|---|---|
| ResNet18 | BadNet | 59.40 / 99.83 | 59.04 / 0.46 | 59.14 / 32.70 |
|  | ISSBA | 59.06 / 99.82 | 57.20 / 0.04 | 59.84 / 82.16 |
| VGG16 | BadNet | 52.52 / 98.51 | 51.54 / 0.18 | 52.52 / 27.00 |
|  | ISSBA | 52.62 / 99.59 | 51.18 / 0.12 | 53.00 / 71.51 |
| EfficientNet-B0 | BadNet | 59.06 / 99.82 | 59.26 / 0.34 | 59.44 / 0.04 |
|  | ISSBA | 58.96 / 99.62 | 56.90 / 0.26 | 58.64 / 16.52 |

Overall, we observe that the results on Tiny ImageNet follow a similar pattern to those on CIFAR10, demonstrating the consistency of our approach across different datasets. However, the ASR values are generally lower compared to CIFAR10, which we attribute to the increased complexity of the Tiny ImageNet dataset. This complexity may make it more challenging for the backdoor to be effectively concealed and subsequently reactivated. Interestingly, we note a unique case with EfficientNet-B0 using the BadNet attack. After fine-tuning, the ASR drops to 0%, which appears to indicate a complete failure of the backdoor activation. However, when we conducted additional experiments with k = 0 (i.e., fine-tuning all layers), we observed an ASR of near 30%. This suggests that the effectiveness of DABF can vary significantly across different model architectures, highlighting the need for tailored strategies in future research to optimize backdoor activation for specific model-attack combinations.

To further understand the behavior of DABF on Tiny ImageNet, we analyzed the impact of varying numbers of fine-tuned layers, as shown in Figure 4. Unlike CIFAR10, where ASR generally increased with fewer fine-tuned layers, Tiny ImageNet shows more diverse patterns. Several models, including ResNet18 BadNet, VGG16 BadNet, and EfficientNet, exhibited inconsistent ASR improvements when fine-tuning only the later layers. This behavior is particularly pronounced in EfficientNet with BadNet attack, where fine-tuning only the last few layers resulted in minimal ASR improvement.

Despite these variations, `DeferBad` demonstrated successful backdoor activation across multiple fine-tuning scenarios, albeit with lower ASR compared to CIFAR10. These results highlight not only the effectiveness of our approach across different datasets but also the complex relationship between model architecture, dataset complexity, and fine-tuning strategies in backdoor activation.

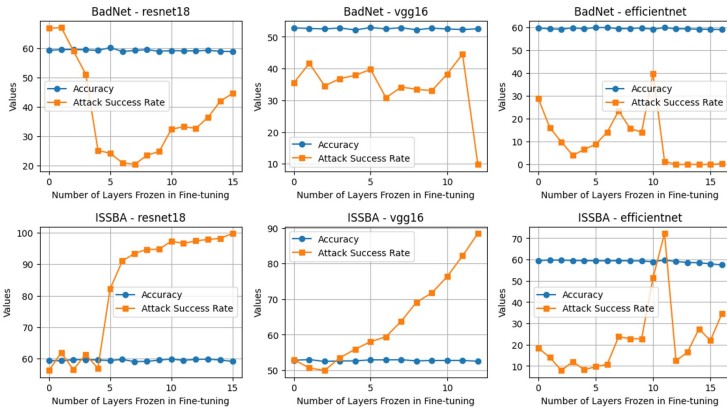

Figure 4: Impact of the number of fine-tuned layers on Clean Accuracy (CA) and Attack Success Rate (ASR) for ResNet18 on Tiny-ImageNet.

Table 6: Clean Accuracy (CA) and Attack Success Rate (ASR) for different models and attack types on CIFAR10-C dataset across corruption types and severities, before and after fine-tuning (FT). Values represent $^{\text{Clean Accuracy (CA)}}/_{\text{Attack Success Rate (ASR)}}$ in percentage.

(a) Gaussian Noise Corruption

| Model | Attack | Severity 1 | | Severity 3 | | Severity 5 | |
|---|---|---|---|---|---|---|---|
| | | Before FT $^{(\uparrow)}/_{(\downarrow)}$ | After FT $^{(\uparrow)}/_{(\uparrow)}$ | Before FT $^{(\uparrow)}/_{(\downarrow)}$ | After FT $^{(\uparrow)}/_{(\uparrow)}$ | Before FT $^{(\uparrow)}/_{(\downarrow)}$ | After FT $^{(\uparrow)}/_{(\uparrow)}$ |
| ResNet18 | BadNet | 80.99 / 0.20 | 90.65 / 86.39 | 46.79 / 0.23 | 81.43 / 72.09 | 34.30 / 0.09 | 77.65 / 64.47 |
| | ISSBA | 77.1 / 1.29 | 90.50 / **91.41** | 40.26 / 1.50 | 80.73 / **91.03** | 29.28 / 1.16 | 76.30 / **91.78** |
| VGG16 | BadNet | 77.65 / 0.01 | 83.7 / **98.20** | 54.16 / 0.0 | 67.88 / **98.12** | 43.91 / 0.01 | 58.84 / **98.69** |
| | ISSBA | 77.10 / 1.29 | 90.50 / **91.41** | 40.26 / 1.50 | 80.73 / **91.03** | 29.28 / 1.16 | 76.3 / **91.78** |
| EfficientNet-B0 | BadNet | 68.14 / 0.85 | 83.44 / 51.56 | 34.36 / 0.43 | 72.18 / 25.29 | 27.31 / 0.58 | 65.08 / 31.46 |
| | ISSBA | 67.96 / 0.39 | 82.93 / **59.50** | 36.54 / 0.15 | 71.09 / **63.84** | 28.84 / 0.08 | 64.73 / **67.48** |

(b) Fog Corruption

| Model | Attack | Severity 1 | | Severity 3 | | Severity 5 | |
|---|---|---|---|---|---|---|---|
| | | Before FT $^{(\uparrow)}/_{(\downarrow)}$ | After FT $^{(\uparrow)}/_{(\uparrow)}$ | Before FT $^{(\uparrow)}/_{(\downarrow)}$ | After FT $^{(\uparrow)}/_{(\uparrow)}$ | Before FT $^{(\uparrow)}/_{(\downarrow)}$ | After FT $^{(\uparrow)}/_{(\uparrow)}$ |
| ResNet18 | BadNet | 93.95 / 0.18 | 95.29 / **97.22** | 91.94 / 0.42 | 94.50 / 89.15 | 75.38 / 0.91 | 89.15 / 79.65 |
| | ISSBA | 94.05 / 0.49 | 94.98 / **89.66** | 91.69 / 0.75 | 94.20 / **93.84** | 75.90 / 1.34 | 89.19 / **90.16** |
| VGG16 | BadNet | 87.85 / 0.00 | 90.20 / **99.00** | 83.71 / 0.00 | 86.91 / **98.93** | 61.61 / 0.00 | 73.74 / 54.07 |
| | ISSBA | 90.23 / 0.70 | 89.59 / **95.74** | 85.53 / 0.98 | 86.98 / **97.55** | 66.08 / 1.10 | 73.79 / **99.76** |
| EfficientNet-B0 | BadNet | 90.65 / 0.57 | 89.95 / 62.64 | 86.34 / 0.84 | 88.45 / 44.09 | 66.36 / 1.12 | 80.78 / 30.21 |
| | ISSBA | 90.06 / 0.40 | 89.19 / **68.62** | 85.68 / 0.44 | 87.40 / **58.57** | 64.29 / 0.31 | 79.49 / 37.65 |

# B ADDITIONAL RESULTS ON CORRUPTED DATASETS: CIFAR10-C, TINY IMAGENET-C

Tables 6 and 7 show the Clean Accuracy (CA) and Attack Success Rate (ASR) for different models and attack types on CIFAR10-C and Tiny ImageNet-C (Hendrycks & Dietterich, 2019). These results encompass various corruption types (Noise, Blur, and Fog) and severity levels.

In CIFAR10-C, our backdoor maintains its effectiveness across different corruption types and severities. Notably, VGG16 exhibits particularly interesting behavior, where the ASR under distribution shift significantly exceeds its performance on the original distribution. For instance, under Gaussian noise corruption, the ASR reaches up to 99.76% (compared to 48.54% on clean data), suggesting that distribution shifts might actually enhance backdoor effectiveness in certain model architectures.

The results on Tiny ImageNet-C reveal even more dramatic patterns. ResNet18 shows remarkably increased ASR under corruption compared to the uncorrupted dataset, achieving over 95% ASR across multiple corruption types and severities (compared to 32.70% on clean data). However, we observe a striking contrast with VGG16 under the BadNet attack, where the ASR drops to nearly 0% after fine-tuning across all corruption types and severities. This stark difference in behavior between architectures highlights the complex interplay between model architecture, dataset complexity, and distribution shifts in backdoor attacks.

Table 7: Clean Accuracy (CA) and Attack Success Rate (ASR) for different models and attack types on TinyImagenet-C dataset across corruption types and severities, before and after fine-tuning (FT). Values represent $^{\text{Clean Accuracy (CA)}}/_{\text{Attack Success Rate (ASR)}}$ in percentage.

(a) JPEG compression Corruption

| Model | Attack | Severity 1 | | Severity 3 | | Severity 5 | |
|---|---|---|---|---|---|---|---|
| | | Before FT (↑)/(↓) | After FT (↑)/(↑) | Before FT (↑)/(↓) | After FT (↑)/(↑) | Before FT (↑)/(↓) | After FT (↑)/(↑) |
| ResNet18 | BadNet | 32.58 / 6.76 | 58.24 / **97.41** | 31.15 / 7.19 | 55.15 / **96.21** | 27.12 / 8.09 | 49.25 / **96.37** |
| | ISSBA | 29.16 / 0.65 | 55.50 / **96.14** | 27.64 / 0.64 | 52.71 / **95.43** | 23.85 / 0.65 | 46.36 / **95.49** |
| VGG16 | BadNet | 29.54 / 0.03 | 40.27 / 0.00 | 29.00 / 0.03 | 38.86 / 0.00 | 26.38 / 0.08 | 36.27 / 0.00 |
| | ISSBA | 29.15 / 1.61 | 41.61 / **77.90** | 28.54 / 1.71 | 39.25 / **77.06** | 25.38 / 2.13 | 35.08 / **74.52** |
| EfficientNet-B0 | BadNet | 33.69 / 7.90 | 55.73 / **3.13** | 33.26 / 8.37 | 53.70 / **2.34** | 29.14 / 9.51 | 48.98 / **0.72** |
| | ISSBA | 32.21 / 4.32 | 55.80 / **16.55** | 32.21 / 4.62 | 55.43 / **18.57** | 27.86 / 4.85 | 48.81 / 7.86 |

(b) Gaussian Noise Corruption

| Model | Attack | Severity 1 | | Severity 3 | | Severity 5 | |
|---|---|---|---|---|---|---|---|
| | | Before FT (↑)/(↓) | After FT (↑)/(↑) | Before FT (↑)/(↓) | After FT (↑)/(↑) | Before FT (↑)/(↓) | After FT (↑)/(↑) |
| ResNet18 | BadNet | 33.39 / 6.27 | 59.12 / **94.64** | 8.43 / 1.55 | 45.11 / **83.40** | 2.66 / 0.45 | 36.80 / **84.23** |
| | ISSBA | 32.27 / 0.48 | 56.70 / **95.20** | 11.29 / 0.14 | 42.10 / 77.14 | 4.45 / 0.06 | 33.45 / 57.75 |
| VGG16 | BadNet | 29.85 / 0.04 | 42.40 / 0.00 | 10.00 / 0.08 | 26.44 / 0.00 | 4.19 / 0.15 | 16.19 / 0.00 |
| | ISSBA | 29.69 / 0.71 | 41.02 / **74.32** | 9.97 / 0.35 | 25.84 / **77.14** | 3.80 / 0.12 | 16.46 / 54.91 |
| EfficientNet-B0 | BadNet | 34.25 / 7.82 | 57.34 / **2.55** | 11.32 / 5.50 | 42.54 / **11.50** | 4.71 / 2.36 | 33.42 / **1.72** |
| | ISSBA | 33.26 / 3.14 | 57.06 / **31.45** | 12.13 / 1.22 | 41.70 / **42.46** | 5.49 / 0.71 | 32.66 / **36.99** |

(c) Fog Corruption

| Model | Attack | Severity 1 | | Severity 3 | | Severity 5 | |
|---|---|---|---|---|---|---|---|
| | | Before FT (↑)/(↓) | After FT (↑)/(↑) | Before FT (↑)/(↓) | After FT (↑)/(↑) | Before FT (↑)/(↓) | After FT (↑)/(↑) |
| ResNet18 | BadNet | 32.49 / 11.54 | 59.62 / **94.16** | 21.41 / 18.44 | 52.94 / **89.58** | 7.06 / 21.01 | 40.84 / **56.66** |
| | ISSBA | 28.32 / 0.83 | 56.49 / **97.51** | 17.62 / 1.28 | 51.64 / **95.47** | 5.56 / 1.05 | 38.24 / **97.48** |
| VGG16 | BadNet | 28.59 / 0.01 | 39.39 / 0.00 | 18.99 / 0.03 | 33.25 / 0.00 | 6.17 / 0.01 | 17.99 / 0.00 |
| | ISSBA | 28.86 / 1.21 | 41.05 / **80.63** | 18.81 / 1.34 | 32.96 / **84.75** | 6.31 / 1.43 | 17.29 / 65.20 |
| EfficientNet-B0 | BadNet | 32.52 / 10.61 | 57.42 / **11.85** | 21.31 / 14.94 | 52.14 / **2.44** | 6.70 / 14.63 | 39.11 / **0.28** |
| | ISSBA | 31.26 / 9.31 | 56.20 / 15.27 | 18.78 / 17.13 | 51.61 / 7.00 | 5.85 / 18.23 | 39.13 / 5.81 |

## C    DISCUSSION ON CONCEALMENT MECHANISM

We provide insights into why our concealment mechanism through selective layer updates creates a state that can be effectively disrupted by fine-tuning, based on our empirical observations. Our experiments suggest that during concealment, the updated layers adapt to counteract backdoor behavior present in other layers. We observe this creates a delicate balance where:

- The updated layers learn parameter values that appear to suppress backdoor signals from other layers

- This suppression represents an unstable solution that differs from natural parameter configurations for the model's primary task

- The concealment effectiveness relies on maintaining specific parameter relationships

When fine-tuning occurs, we observe:

- The optimization process alters these carefully balanced parameters

- This disrupts the suppression mechanism

- The model shifts to a state where backdoor features become active again

While the exact mathematical nature of this mechanism warrants further theoretical investigation, our extensive experiments consistently demonstrate this behavior across different architectures and scenarios.

## D    ADDITIONAL DETECTION METHODS

We evaluated `DeferBad` against three additional state-of-the-art backdoor detection methods: Random Channel Shuffling (RCS), Scale-Up, and IDB-PSC.

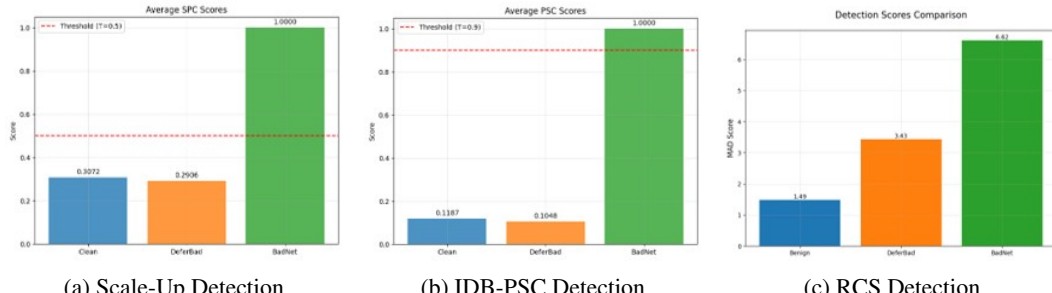

| (a) Scale-Up Detection | (b) IDB-PSC Detection | (c) RCS Detection |

Figure 5: Detection results of `DeferBad` against additional backdoor detection methods. (a) Scale-Up detection shows similar consistency scores between `DeferBad` (0.2906) and benign models (0.3072). (b) IDB-PSC detection demonstrates `DeferBad`'s effectiveness in evading detection with scores (0.1048) close to benign models (0.1187). (c) RCS detection reveals some capability in detecting `DeferBad` (3.43) compared to benign models (1.49), but significantly lower than BadNet (6.62).

**Random Channel Shuffling (RCS):** RCS (Cai et al., 2022) exploits the observation that trigger information tends to be concentrated in specific channels by randomly shuffling channels and observing class-wise variations. Our experiments showed that while RCS could detect `DeferBad` with an anomaly score of 3.43 (compared to 1.49 for benign models), this was significantly lower than the score of 6.62 for conventional BadNet attacks (Figure 5c). This suggests that while `DeferBad` is detectable by RCS, it demonstrates improved stealthiness compared to conventional attacks. Furthermore, this relative improvement indicates potential for future refinements of `DeferBad` to completely evade RCS detection.

**Scale-Up Detection:** Scale-Up (Guo et al., 2023) detects backdoors by examining prediction consistency under image amplification. `DeferBad` successfully evaded this detection method, achieving an SPC score of 0.2906, which is slightly lower than benign models (0.3072) and significantly different from BadNet attacks (1.0), as shown in Figure 5a.

**IDB-PSC Detection:** IDB-PSC (Hou et al., 2024) detects backdoors by analyzing consistency under batch normalization parameter scaling. Our experiments demonstrated that `DeferBad` effectively evaded this detection method, with a score of 0.1048 compared to 0.1187 for benign models and 1.0 for BadNet attacks (Figure 5b).

These additional experiments further validate the stealthiness of `DeferBad` across a broader range of detection methods, particularly showing strong evasion capabilities against Scale-Up and IDB-PSC detection methods.

## E    ANALYSIS OF LATENT BACKDOOR BEHAVIOR

To better understand the differences between our approach and latent backdoors (Yao et al., 2019), we analyzed the behavior of latent backdoors during their dormant phase. Specifically, we examined the model's output distributions for clean and triggered inputs using the PubFigure dataset, where each class has an equal number of samples.

Figure 6 shows the mean and variance of model predictions across different classes for both clean and triggered inputs. For clean inputs, we observe that the model's predictions follow a relatively uniform distribution across classes, which is expected given the balanced nature of the dataset. However, when presented with triggered inputs, the model exhibits anomalous behavior: certain classes show unusually high confidence (high mean) in predictions, while multiple classes display near-zero variance in their prediction distributions. This stark contrast in behavior is particularly suspicious given that the dataset has a uniform class distribution.

This observation reveals a critical weakness in latent backdoors. Even during their dormant phase, they process triggered inputs in a distinctly different manner that manifests in the model's output distributions. The presence of highly confident predictions and unnaturally low variances for certain classes, despite the uniform class distribution in the dataset, creates a clear signal that could be

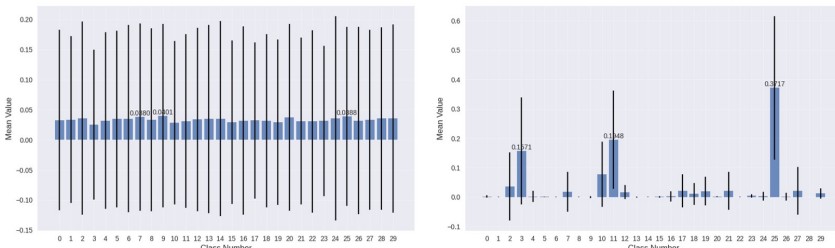

Figure 6: Analysis of model predictions for clean and triggered inputs in a dormant latent backdoor model (Yao et al., 2019) on the PubFigure dataset. Left: For clean inputs, predictions show expected uniformity across classes. Right: For triggered inputs, certain classes exhibit unusually high confidence (mean) and multiple classes show near-zero variance, despite the balanced dataset.

exploited for detection. In contrast, as shown in Figure 7b, `DeferBad` maintains natural output distributions for both clean and triggered inputs during its dormant phase, achieving true concealment of the backdoor.

## F ANALYSIS OF MODEL OUTPUT DISTRIBUTIONS

We analyzed the output distributions of different backdoor approaches during their dormant phase using the CIFAR-10 dataset. Figure 7 shows the mean and variance of model predictions across different classes for both clean and triggered inputs.

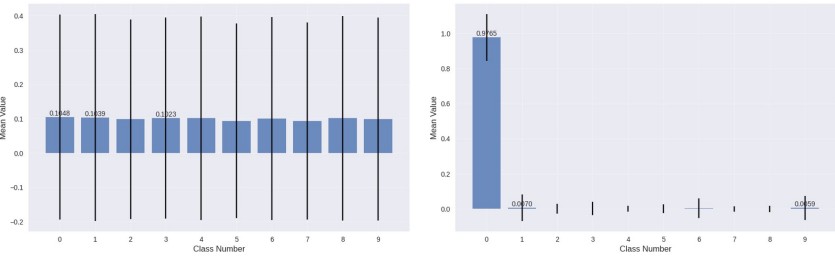

(a) Conventional backdoor (Gu et al., 2017) distinctly different patterns between clean and triggered inputs.

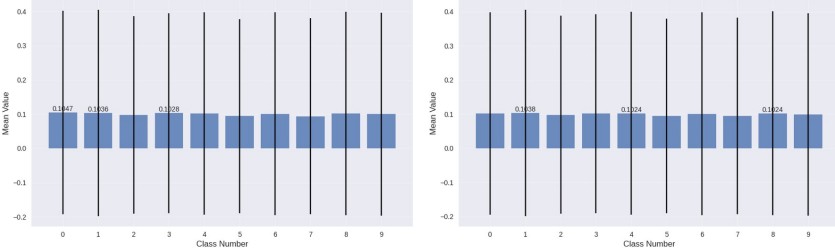

(b) DeferBad exhibits nearly identical distributions between clean and triggered inputs.

Figure 7: Comparison of model output distributions for clean (left) and triggered (right) inputs during the dormant phase. Output distributions are visualized using means and variances across classes.

As shown in the figure, conventional backdoors (Gu et al., 2017) produce noticeably different output patterns when presented with triggered inputs, making them potentially detectable through output distribution analysis. In contrast, `DeferBad` maintains virtually indistinguishable output distributions between clean and triggered inputs, successfully concealing the backdoor's presence.

