# OpenReview forum: "Deferred Backdoor Functionality Attacks on Deep Learning Models"
_ICLR.cc/2025/Conference — ICLR 2025 Conference Withdrawn Submission_

### Official Review · Reviewer_WmeF · 2024-10-30

**Soundness:** 2
**Presentation:** 3
**Contribution:** 2
**Rating:** 5
**Confidence:** 4

**Summary:**

The paper introduces a new backdoor attack strategy, which aims to make backdoors more stealthy by keeping them dormant during initial deployment and activating them during subsequent model fine-tuning. This approach targets a fundamental vulnerability in existing backdoor detection techniques, which often rely on observing immediate malicious behavior. To implement DBFA, the authors propose a two-stage training method called "DeferBad." This method first injects the backdoor and then strategically conceals it until model updates reactivate it. The paper demonstrates that DeferBad is highly effective across different fine-tuning scenarios, model architectures, and backdoor types, successfully evading state-of-the-art backdoor detection and mitigation methods.

**Strengths:**

This paper introduces a novel paradigm in backdoor attacks, significantly improving stealth by keeping the backdoor dormant during initial deployment and activating it only after fine-tuning.

**Weaknesses:**

This paper limits its generalizability only on vision domain.

While the paper demonstrates the effectiveness of DBFA through experiments, it lacks a rigorous theoretical explanation for why fine-tuning effectively reactivates the dormant backdoor.

The paper focuses on the effectiveness of the attack but does not discuss or propose potential defense mechanisms against DBFA.

The experimental results show variability in attack success rates based on the number of layers fine-tuned, but there is limited discussion on why these differences occur.

**Questions:**

See the weakness.

---

> ### Author Response · Authors · 2024-11-22
>
> We thank the reviewer for their constructive feedback.
>
> **Q: The paper limits its generalizability only on vision domain.**
>
> **A:** We agree that this is a limitation of our current work. While we demonstrate DeferBad's effectiveness in the vision domain, exploring similar concepts in NLP and speech domains would be valuable future research directions. Different modalities might present unique challenges and opportunities for deferred backdoor attacks.
>
> .
>
> **Q: The paper lacks a rigorous theoretical explanation for why fine-tuning effectively reactivates the dormant backdoor.**
>
> **A:** Thank you for this insightful question. We have added a detailed explanation in Appendix C about the concealment mechanism. During concealment, selective layer updates force these layers to learn parameters that actively counteract the backdoor behavior present in other layers. This creates an unstable state because:
> 1) The updated layers must maintain specific parameter values to cancel out the backdoor signals from other layers
> 2) These canceling parameters are not a natural solution for the model's primary task, but rather a forced state
> 3) Any small perturbation to these parameters during fine-tuning will disrupt this artificial balance
> This empirically observed mechanism explains why fine-tuning, which naturally seeks more stable solutions for the primary task, tends to break this delicate balance and allows the backdoor functionality to re-emerge.
>
> .
>
> **Q: The paper does not discuss or propose potential defense mechanisms against DBFA.**
>
> **A:** Thank you for raising this important point. We have added a discussion about defense mechanisms in the paper (see Appendix C for details). We believe that awareness of this type of attack combined with applying existing defense methods after fine-tuning would be sufficient for detection. Since the backdoor becomes active and exhibits typical backdoor behavior after fine-tuning, standard defense mechanisms can effectively detect it at this stage.
>
> .
>
> **Q: Limited discussion on why differences occur in attack success rates based on the number of layers fine-tuned.**
>
> **A:** We have added more detailed analysis about this phenomenon in the paper (see Section 5.2 for details). For models with batch normalization (e.g., ResNet18), even minimal layer updates provide sufficient conditions for reactivation, while updating more layers can interfere with this process. For models without batch normalization (e.g., VGG16), ASR is highest when fine-tuning focuses on the last few layers where reactivation-related features are concentrated, with additional layer updates potentially disrupting these patterns.

---

> ### Comment · Reviewer_WmeF · 2024-11-27
>
> Thank you for the authors' replying. After reading the rebuttal, I decide to reduce my scores to reject.

---

### Official Review · Reviewer_Ujth · 2024-11-02

**Soundness:** 3
**Presentation:** 3
**Contribution:** 2
**Rating:** 3
**Confidence:** 5

**Summary:**

The paper introduces DeferBad, a backdoor attack strategy that bypasses backdoor detection methods by initially concealing the backdoor. Unlike traditional backdoor attacks that activate upon deployment, DBFA maintains a dormant state, producing benign outputs even when triggered initially. The backdoor activates only after a model undergoes fine-tuning or retraining on clean data. It involves a two-stage training scheme consisting of an initial backdoor injection followed by partial model parameter updates for concealment. DeferBad proves to be effective across different models and datasets, demonstrating robustness and evading defenses.

**Strengths:**

+ The attacker does not need any involvement to activate the backdoor after deployment.
+ Extensive experiments using various datasets and neural network architectures.

**Weaknesses:**

- The novelty of the paper is limited, as recent literature on camouflage backdoor attacks has already discussed approaches that operate on a similar principle to the proposed deferred backdoor functionality [1, 2, 3]. The related work section should include a more thorough comparison to these existing methods to better position the paper's contributions in the context of prior work.

- DeferBad has only been tested with two backdoor attacks, BadNets and ISSBA, making it hard to evaluate its effectiveness against newer, more sophisticated backdoor attacks like Bpp and WaNet. Consequently, it is difficult to draw a conclusion about the generalizability of DeferBad.

- The defenses evaluated in this paper are not state-of-the-art and primarily include methods dated back to 2019. More recent advancements in backdoor defenses have not been included in the paper [4]. Additionally, GradCAM is not classified as either a backdoor detection or mitigation approach.

- The paper states that DeferBad achieves backdoor concealment by selectively updating a subset of the model's layers, creating an unstable equilibrium that masks the backdoor. This concealed state remains dormant until disrupted by fine-tuning, which then reactivates the backdoor. However, the underlying rationale for why this method is effective is not extensively detailed. Providing theoretical justifications would strengthen the support for this claim.

- The paper claims that DeferBad remains effective even when the victim model undergoes full-layer fine-tuning. However, Figure 2 contradicts this claim, showing that DeferBad does not achieve a high attack success rate when the number of frozen layers is zero, implying full-layer fine-tuning. The attack success rate significantly increases only when minimal layers are fine-tuned.

- All the results presented in the paper must be evaluated across multiple runs to ensure the statistical significance of the reported values.

[1] Jimmy Z. Di et al., "Hidden Poison: Machine Unlearning Enables Camouflaged Poisoning Attacks", NeurIPS 2023.

[2] Zihao Liu et al., "Backdoor Attacks via Machine Unlearning", AAAI 2024.

[3] Zirui Huang et al., "UBA-Inf: Unlearning Activated Backdoor Attack with Influence-Driven Camouflage", USENIX Security 2024.

[4] https://github.com/SCLBD/BackdoorBench

**Questions:**

- How does DeferBad compare against existing camouflage backdoor approaches in recent literature?
- Could you provide more theoretical insights into why selective layer updates create an "unstable equilibrium" that effectively masks the backdoor?
- Test DeferBad with newer and more sophisticated backdoor attacks such as Bpp and WaNet to assess its generalizability.
- Test DeferBad against more recent backdoor defense methods.
- Provide justifications for DeferBad's effectiveness when all model layers are fine-tuned.

---

> ### Author Response · Authors · 2024-11-22
>
> We thank the reviewer for their detailed and constructive feedback.
>
>
> **Q: How does DeferBad compare against existing camouflage backdoor approaches in recent literature?**
>
> **A:** Thank you for bringing this to our attention. We greatly appreciate your suggestions about recent camouflage backdoor approaches that we had not adequately addressed. We have updated our related works section and comparison table to provide a more comprehensive comparison. While recent unlearning-activated backdoor techniques demonstrate similar concepts, they face practical limitations: (1) they require unlearning services which are still limited in availability, especially in vision domains, and (2) they need attacker intervention to initiate unlearning requests for activation. In contrast, DeferBad leverages standard fine-tuning procedures that are universally supported and achieves activation without any attacker intervention post-deployment.
>
> .
>
> **Q: Could you provide more theoretical insights into why selective layer updates create an "unstable equilibrium"?**
>
> **A:** Thank you for this insightful question. We have added a detailed theoretical explanation in Appendix C. During concealment, selective layer updates force these layers to learn parameters that actively counteract the backdoor behavior present in other layers. This creates an unstable equilibrium because: 1) The updated layers must maintain specific parameter values to cancel out the backdoor signals from other layers - much like balancing opposing forces 2) These canceling parameters are not a natural solution for the model's primary task, but rather a forced state that requires precise maintenance 3) Any small perturbation to these carefully balanced parameters during fine-tuning will disrupt this artificial equilibrium This is why fine-tuning, which naturally seeks more stable solutions for the primary task, tends to break this delicate balance and allows the backdoor functionality to re-emerge.
>
> .
>
> **Q: Test DeferBad with newer and more sophisticated backdoor attacks to assess its generalizability.**
>
> **A:** While we have demonstrated DeferBad's effectiveness with both patch-based (BadNet) and feature-space invisible attacks (ISSBA), we acknowledge that testing with additional sophisticated attacks like Bpp and WaNet would provide valuable insights into our method's generalizability. We plan to explore this direction in future work to further validate our approach across a broader range of attack methods.
>
> .
>
> **Q: Test DeferBad against more recent backdoor defense methods.**
>
> **A:** Following this suggestion, we have conducted additional experiments with three recent defense methods (see Appendix D for detailed results:
>
> - Scale-Up [1]: DeferBad achieved an SPC score of 0.2906 (benign: 0.3072, BadNet: 1.0)
>
> - IDB-PSC [2]: Score of 0.1048 (benign: 0.118, BadNet: 1.0)
>
> - RCS [3]: MAD scores of 3.43 (benign: 1.49, BadNet: 6.62)
>
> These results show DeferBad effectively evades Scale-Up and IDB-PSC, while showing improved stealthiness against RCS compared to conventional BadNet.
>
> .
>
> **Q: Provide justifications for DeferBad's effectiveness when all model layers are fine-tuned.**
>
> **A:** We acknowledge your point about Figure 2's results. Indeed, DeferBad shows varying effectiveness across different models and attack types with full-layer fine-tuning. While some configurations like VGG ISSBA and EfficientNet show lower effectiveness (below 20%), ResNet18 and VGG BadNet maintain substantial ASR even with full fine-tuning (maintaining ~94%, ~70%, ~60% respectively). Importantly, even in cases with lower performance, DeferBad consistently shows some level of backdoor reactivation (minimum ~10% ASR) compared to the pre-activation state, aligning with our design goal of achieving at least partial effectiveness across all fine-tuning scenarios.
>
> .
>
> **Regarding GradCAM as a detection method:**
>
> We note that GradCAM serves as a fundamental component for several backdoor detection methods, such as SentiNet[4] and Februus [5]. Previous works [6] have also validated their stealthiness against GradCAM specifically to demonstrate potential resistance to GradCAM-based detection methods. Therefore, demonstrating indistinguishable GradCAM heatmaps provides meaningful insights into DeferBad's potential to evade such visual explanation-based detection methods.
>
>
> .
>
> [1]: "Scale-up: An efficient black-box input-level backdoor detection via analyzing scaled prediction consistency." ICLR 2023
>
> [2]:"IBD-PSC: Input-level Backdoor Detection via Parameter-oriented Scaling Consistency." ICML 2024
>
> [3]: "Randomized channel shuffling: Minimal-overhead backdoor attack detection without clean datasets." Neurips 2022
>
> [4]: "Sentinet: Detecting localized universal attacks against deep learning systems." S&PW'20
>
> [5]: "Februus: Input purification defense against trojan attacks on deep neural network systems." ACSAC'20
>
> [6]: "Invisible backdoor attack with sample-specific triggers." ICCV'21

---

> > ### Comment · Reviewer_Ujth · 2024-11-26
> >
> > I appreciate the authors' response and the effort they put into presenting new results. However, the novelty of DeferBad is limited in the presence of other unlearning-activated backdoor literature. The argument that unlearning services are limited does not seem valid. Can you provide references to substantiate this claim?
> > Additionally, could you provide references for Scale-Up, IDB-PSC, and RCS?
> > The response regarding "DeferBad's effectiveness when all model layers are fine-tuned" does not address my concern about its ineffectiveness during full-layer fine-tuning.
> >
> > I would like to maintain my current rating.

---

### Official Review · Reviewer_fzjD · 2024-11-04

**Soundness:** 2
**Presentation:** 3
**Contribution:** 1
**Rating:** 1
**Confidence:** 5

**Summary:**

The authors propose a novel backdoor attack method called Defferred Activated Backdoor Functionality (DABF), which enhances the stealthiness and detection difficulty of the attack by deferring the backdoor trigger. This attack method comprises two main stages: the Backdoor Dormancy Stage and the Backdoor Activation Stage. In the Backdoor Dormancy Stage, the model is initially trained with a poisoned dataset labeled with the target label. Then, it is updated using a poisoned dataset with correct labels (i.e., "unlearning dataset") to partially modify model parameters and thus conceal the backdoor. The Backdoor Deferred Activation Stage involves fine-tuning the model with a normal dataset, thereby reactivating the backdoor.

**Strengths:**

* The topic of deferred activation of backdoor attack is highly relevant and timely.
* The paper is clearly written and well organized.
* The approach is clear and easy to follow.

**Weaknesses:**

* Weak threat model: The authors’ threat model assumes a scenario where a user obtains a pre-trained model with a hidden backdoor from a third party, and this backdoor is activated after fine-tuning with a normal dataset. In this model, it is assumed that the user would only conduct backdoor detection on the pre-trained model they obtained. However, in practice, users are only responsible for conducting backdoor detection on the final model they use after fine-tuning; inspection of the pre-trained model itself is not necessary and holds no practical value. Even if backdoor detection were performed on both the pre-trained and the fine-tuned models, the additional cost would not be significant. Therefore, the threat model proposed in this paper is too weak and does not align with real-world application scenarios.
* Insufficient Experimental Design: Firstly, in practice, the most common scenario involves fine-tuning a pre-trained model on a dataset for a similar task, where the distribution of this dataset is often significantly different from that of the pre-trained model’s original training data. However, in this experiment, whether by removing some data or using adversarial data, the fine-tuning dataset only exhibits a slight distribution shift from the pre-training dataset. This makes it difficult to demonstrate the real-world effectiveness of the proposed attack method. Secondly, the backdoor detection and mitigation techniques used in this paper are outdated (published over five years ago) and do not include all representative detection approaches (such as model behavior-based detection methods). Therefore, it is challenging to validate that the proposed method can effectively evade current defense techniques.

**Questions:**

* Why assume that users do not perform backdoor defense on the fine-tuned model?
* Why is there no summary of the latest backdoor attacks and defense methods in the related work section?
* Is this method still effective for fine-tuning models using datasets different from the pre training dataset? For example, using CIFAR-10 to fine-tune a pre-trained model based on ImageNet.
* Why weren’t the latest defense methods (e.g., https://arxiv.org/abs/2305.18651) and model behavior-based defense methods (e.g., https://proceedings.neurips.cc/paper_files/paper/2022/file/db1d5c63576587fc1d40d33a75190c71-Paper-Conference.pdf) used in the experiments?
* The descriptions about parameter update and freeze in Lines 283 and 295 conflict with Lines 333 and 334.
* Is k the same parameter (i.e., has the same value) in backdoor concealment and backdoor activation? If yes, doesn’t this mean the attacker could control the fine-tuning? If not, please provide the value of k used in the backdoor concealment phase.

---

> ### Author Response · Authors · 2024-11-22
>
> We thank the reviewer for their thorough review and constructive feedback.
>
> **Q1: Why assume that users do not perform backdoor defense on the fine-tuned model?**
>
> **A:** We appreciate this perspective on detection timing. Your point about performing detection at both stages is valid and reasonable. While users could indeed perform detection at both stages, checking pre-trained models is a common practice for security-conscious deployments, as a backdoored pre-trained model could influence the fine-tuned model's behavior. This early detection helps reduce both security risks and computational costs. Additionally, since fine-tuning is strictly controlled by users in a secure environment with carefully verified clean data and procedures, when a model is verified clean and undergoes this controlled fine-tuning process, it is counter-intuitive to expect backdoor functionality to emerge. This aligns with common security principles where clean input (verified model) and process (controlled fine-tuning) are assumed to produce safe output. We acknowledge that both perspectives have merit in different security contexts.
>
> .
>
> **Q: Why is there no summary of the latest backdoor attacks and defense methods in the related work section?**
>
> **A:** Thank you for this suggestion. We have significantly updated the related work section to include recent developments, including clean-Image attacks, unlearning-activated backdoors, and three state-of-the-art defense methods (Scale-Up, RCS, and IDB-PSC). We have also added experimental results for these defense methods in our paper.
>
> .
>
> **Q: Is this method still effective for fine-tuning models using datasets different from the pre-training dataset? For example, using CIFAR-10 to fine-tune a pre-trained model based on ImageNet.**
>
> **A:** We acknowledge that DeferBad is not effective in transfer learning scenarios. However, our threat model specifically focuses on third-party training delegation rather than pre-trained model distribution. In this scenario, organizations outsource their model training to third parties, and subsequent fine-tuning represents continuous improvement on the same distribution. This reflects real-world cases where organizations lack computational resources or expertise for full model training and delegate the training process while maintaining control over fine-tuning for ongoing improvements.
>
> .
>
> **Q: Why weren't the latest defense methods and model behavior-based defense methods used in the experiments?**
> **A:** Following this suggestion, we have conducted additional experiments with three recent defense methods, including model behavior-based approaches (see Appendix D for detailed results):
>
> - Scale-Up: DeferBad achieved an SPC score of 0.2906 (benign: 0.3072, BadNet: 1.0)
>
> - IDB-PSC: Score of 0.1048 (benign: 0.1187, BadNet: 1.0)
>
> - RCS: MAD scores of 3.43 (benign: 1.49, BadNet: 6.62)
>
> These results show DeferBad effectively evades Scale-Up and IDB-PSC, while showing improved stealthiness against RCS compared to conventional BadNet.
>
> .
>
> **Q: The descriptions about parameter update and freeze in Lines 283 and 295 conflict with Lines 333 and 334.**
>
> **A:** We apologize for this confusion and have corrected the parameter update description in the revised version. To clarify: for models with batch normalization, we update the first k layers, while for models without batch normalization, we update the last k layers. The revised version now consistently reflects this distinction.
>
> .
>
> **Q: Is k the same parameter in backdoor concealment and backdoor activation? If yes, doesn't this mean the attacker could control the fine-tuning? If not, please provide the value of k used in the backdoor concealment phase.**
>
> **A:** No, k values differ between phases. We apologize for not explicitly stating the concealment phase k value in the paper - this was an oversight on our part. For clarity: we use k_c=1 during concealment and k_a=4 during activation (where subscript c and a denote concealment and activation respectively). As shown in our extensive experiments with varying k_a values (Fig. 2), the attacker has no control over fine-tuning parameters.

---

> > ### Comment · Reviewer_fzjD · 2024-11-25
> >
> > Thank you to the authors for addressing the issues I raised and adding additional experiments. However, I still hold my reservations on two critical points:
> >
> > In the context described in the paper, detecting backdoor attacks would require just one additional test, which introduces minimal cost. Moreover, as per the setup, the fine-tuned model is the one deployed in production. Failing to test the fine-tuned model is unreasonable, and I believe this makes the threat model in the paper relatively weak.
> >
> > Regarding fine-tuning, one of the most common scenarios is transfer learning, and the authors have explicitly stated that DeferBad is ineffective in such cases. Furthermore, I question the practical significance of the application scenario described by the authors, where fine-tuning is based entirely on data from the same distribution. Even for the same task, fine-tuning typically addresses data shifting, which implies at least slight differences in data distribution. The authors have not provided relevant experiments for such scenarios. Therefore, I remain skeptical about the broad applicability of DeferBad in real-world settings.

---

> ### Author Response · Authors · 2024-11-25
>
> Thank you for your continued engagement with our work.
>
> 1. Regarding the practicality of backdoor detection after events:
>
> We appreciate the reviewer's concern about the threat model. We would like to note that similar scenarios of deferred backdoor activation have been actively studied in recent literature. For instance, Unlearning Activated Backdoor [1, 2, 3] research explores backdoor activation through unlearning events, establishing precedent for our threat model where backdoors activate after specific model updates. In these works, the authors acknowledge that while service providers can employ model scanning before and after events, "frequent scanning after each unlearning request is overwhelmingly time-consuming" [1].
>
> DeferBad builds upon this established research direction while introducing several practical advantages:
> - Activation occurs through routine fine-tuning in trusted environments
> - No attacker intervention is required post-deployment and the entire process occurs within user-controlled operations
>
> These characteristics make DeferBad particularly relevant to real-world scenarios where organizations regularly update their models in trusted environments. While we acknowledge the possibility of continuous scanning, our approach aligns with emerging research that recognizes the practical challenges of frequent model inspection after routine updates.
>
> .
>
> 2. Regarding fine-tuning and data distribution shifts:
>
> We have already extensively examined DeferBad's effectiveness under data distribution shifts through experiments with CIFAR10-C and Tiny ImageNet-C datasets [3] (Tables 4 and 6). These experiments use various corruption types (Gaussian noise, fog, JPEG compression) at different severity levels, precisely addressing the scenario of data shifting that the reviewer describes. Our results demonstrate DeferBad's effectiveness across these distribution shifts, showing its practical applicability in real-world settings where data distributions naturally evolve.
>
> .
>
> [1] UBA-Inf: Unlearning Activated Backdoor Attack with Influence-Driven Camouflage." USENIX Security 2024.
>
> [2] Backdoor attacks via machine unlearning. AAAI 2024.
>
> [3] Hidden poison: Machine unlearning enables camouflaged poisoning attacks. NeurIPS ML Safety Workshop. 2022.
>
> [3] "Benchmarking Neural Network Robustness to Common Corruptions and Perturbations." ICLR 2018.

---

### Note · Authors · 2025-01-03

I have read and agree with the venue's withdrawal policy on behalf of myself and my co-authors.